# Consumer Perception and Liking of Parmigiano Reggiano Protected Designation of Origin (PDO) Cheese Produced with Milk from Cows Fed Fresh Forage vs. Dry Hay

**DOI:** 10.3390/foods13020309

**Published:** 2024-01-18

**Authors:** Matilde Tura, Mara Antonia Gagliano, Francesca Soglia, Alessandra Bendini, Francesca Patrignani, Massimiliano Petracci, Tullia Gallina Toschi, Enrico Valli

**Affiliations:** 1Department of Agricultural and Food Sciences, Alma Mater Studiorum—Università di Bologna, Viale Fanin 40, 40125 Bologna, Italy; matilde.tura2@unibo.it (M.T.); tullia.gallinatoschi@unibo.it (T.G.T.); 2Interdepartmental Centre of Industrial Agrifood Research, Alma Mater Studiorum—Università di Bologna, Via Quinto Bucci 336, 47521 Cesena, Italy; francesca.soglia2@unibo.it (F.S.); francesca.patrignani@unibo.it (F.P.); m.petracci@unibo.it (M.P.); enrico.valli4@unibo.it (E.V.); 3Department of Agricultural and Food Sciences, Alma Mater Studiorum—Università di Bologna, Piazza Goidanich 60, 47521 Cesena, Italy; maraantonia.gagliano@unibo.it

**Keywords:** cheese, protected designation of origin, feed, sensory analysis, color, image analysis, consumers, check-all-that-apply

## Abstract

This study aimed to investigate consumer sensory profiles and liking of Parmigiano Reggiano PDO cheese produced with milk from cows reared indoors and fed with different forage sources, i.e., dry hay and fresh forage. Two cheese samples were tested by 119 Italian subjects, following a protocol that included a Check-All-That-Apply method to assess the sensory profile, a Just-About-Right scale to evaluate the adequacy of attributes, and questions on liking (9-point hedonic scale). A questionnaire related to personal information and consumption habits was also submitted. The color of the two samples, based on image analysis, was different: the sample produced with milk from the dairy cows fed fresh forage had a higher intensity of yellow than the other; they were also described differently (*p* ≤ 0.05) by participants in the consumer test. Indeed, Parmigiano Reggiano produced with milk from the cows that were fed dry hay was mainly characterized by a “*fresh milk*” and “*solubility*”, while the sample produced with milk from cows fed fresh forage was described as “*yellow*”, “*seasoned*”, “*pungent*”, and with a “*cheese crust*” flavor. Even if no significant differences were observed between the two samples in terms of liking (*p* ≤ 0.05), the attribute “*graininess*” showed a great impact on liking ratings together with “*yellow*” (*p* ≤ 0.05), apparently corresponding to a specific expectation regarding the intensity of these attributes. Data were also analyzed according to the gender of consumers, highlighting that for women, the adequacy of “*fresh milk”*, “*sweet”*, and “*graininess”* greatly impacted liking for the cheese from cows fed dry hay.

## 1. Introduction

Nowadays, consumers are increasingly aware of the quality of dairy products and also pay increased attention to the quality of the milk from which these products are produced, along with its geographical origin [1]. Dairy products with a protected designation of origin (PDO) are characterized by a deep connection between the geographical area of their production, dairy cow feed, human elements, and knowledge about their quality and characteristics. Among the PDO cheeses available on the market, Parmigiano Reggiano is one of the most widely exported, consumed, and known worldwide [2,3]. This PDO product is an extra-hard Italian cheese produced in a specific area of the North of Italy, manufactured from raw bovine milk [4,5].

Its production process complies with a strict standard, which also regards the composition of the diet for dairy cows [6,7], and its production is supervised by the Parmigiano Reggiano Cheese Consortium. Moreover, the designation “PDO Parmigiano Reggiano” can be attributed to cheese produced with traditional methods in a specific zone of Italy (provinces of Parma, Reggio Emilia, Modena, Mantova, and Bologna) from milk produced in the same area [8,9,10]. In particular, the milk has to be partially skimmed by natural creaming and poured into a traditional open copper vat with natural whey starter and calf rennet to achieve coagulation (33–34 °C, 8–10 min). The curd needs to be cut into rice grain-sized granules, and the temperature is progressively raised to 53–54 °C in 10–12 min to allow sedimentation and aggregation. Next, the curd is extracted, cut into two portions, and placed into a circular mold. The cheese wheels are salted in brine for 20–22 days and then ripened (18 °C and 80% relative humidity). The minimum seasoning period for PDO Parmigiano Reggiano is 12 months [5].

In recent years, dairy products produced from pasturing or grazing have received more interest by consumers. This increased attention may be linked to increased consumer awareness of the health benefits of dairy fats (conjugated linoleic acid) associated with these farming systems, as well as food authenticity, environmental sustainability, animal-welfare (less-intensive farming), local origin, geographical origin, and organic production systems [11]. Several factors can influence the characteristics of milk and cheese, such as animal breed, age, health status, stage of lactation, feeding regimen, and seasonality [12,13]. However, the impact of the cow’s diet on consumers’ sensory perceptions of bovine milk and dairy products is unclear because a wide range of potential factors are involved at the farm level (e.g., type of forage, grassland management, forage conservation methods/practices, animal breeds, stage of lactation, health status of the animal) and during production (e.g., heat treatment, product type, storage fermentation) [11]. Several studies have highlighted that forage in dairy cattle diets can modify the characteristics of milk and cheese, also in relation to their sensory profile [4,12,14,15,16,17]. Different pastures are recognized to affect the composition of milk, conferring specific organoleptic traits to the milk in comparison to silage- or cereal-based concentrate feed [1,18]. Such a link between the type of forage consumed by dairy cows and their milk’s chemical composition results from the ability of several plant components to be transferred to milk and eventually to cheese through a carry-over process [19].

To the authors’ knowledge, consumer perception of sensory aspects and consumer liking of Parmigiano Reggiano PDO in relation to changes in the type of cow feed have not been investigated widely in the literature. Considering that the quality of the product is closely linked to the PDO specification sheet, outlined in regulations, this study aims to investigate the differences in sensory characteristics perceivable by consumers in relation to variations in the cows’ diet. Thus, the aim of this research is to assess the sensory profile and liking by Italian consumers of two different Parmigiano Reggiano PDO cheeses, comparing the features of cheese manufactured with milk from dairy cows farmed indoors and fed dry hay with those of cheese produced from the milk of animals fed fresh forage.

## 2. Materials and Methods

### 2.1. Ethical Approval 

This study was approved by the Alma Mater Studiorum—Università di Bologna Bioethical Committee (prot. no. 0057914, date 3 March 2023).

### 2.2. Samples

Two batches of Parmigiano Reggiano PDO cheese, manufactured with milk from dairy cows farmed indoors and fed the same diet except for the forage source (i.e., dry hay vs. fresh forage), were considered to evaluate consumer liking and to highlight possible perceived differences related to the cow feed. The two batches were coded as P-DG (Parmigiano Reggiano from cows fed 15–18 kg of dry hay/cow/day) and P-FG (Parmigiano Reggiano from cows fed 40 kg of fresh forage/cow/day plus 9–12 kg of dry hay). Both batches were produced during the summer of 2021 in the same dairy (located in the lowland area of Azienda Agricola Ciaolatte, Italy, with GIS coordinates of 44°49′17.94′’ N 10° 6′29.416′’ E) and shipped by the Consortium of Parmigiano Reggiano PDO—“Consorzio del Formaggio Parmigiano Reggiano” (Reggio Emilia, Italy)—to the University of Bologna facilities (Campus of Food Science, Cesena, Italy) for sensory and instrumental analyses. The farming and production processes followed a single document [6]. Both batches were sourced from organic agricultural practices and seasoned for 24 months. Each sample consisted of a single piece of cheese from the same wheel, produced from a bulk quantity of milk to simulate commercial conditions, for a total of 2 kg per batch. 

Prior to the analysis, samples were stored at 4 °C. Samples to be presented to participants for sensory analysis were obtained by cutting the products into 4 g parallelepipeds; for each sample, 3 pieces were tested.

### 2.3. Participants in the Sensory Test

A total of 119 subjects (aged 18–70 years old, with 86 subjects being between 20 and 25 years old; 68 women, 48 men, 1 non-binary person, and 2 whose gender was not declared; all from Italy, with 74.8% being from the Emilia-Romagna region) participated in the study. Participants were recruited among the students and the staff of the Department of Agricultural and Food Sciences of the Alma Mater Studiorum—Università di Bologna, and participated in the session following the evaluation procedure described in Section 2.4. To meet the inclusion criteria, each subject had to be aged between 18 and 70 years and a resident in Italy, while the exclusion criteria excluded people with lactose intolerance or allergies to milk proteins. Participants provided their informed consent to participate, as well as completing a privacy information sheet. Individuals voluntarily joined the study, which took around 30 min to complete the required tasks.

### 2.4. Evaluation Procedure in the Sensory Test 

For the sensory test, Parmigiano Reggiano samples (parallelepipeds weighing about 4 g) were presented in disposable paper dishes (Appendix A) codified with three-digit random codes, in a randomized and balanced order, at room temperature. All of the cheese products were analyzed over two weeks in April 2023. One repetition was conducted. Consumer evaluation included: (1) a liking test (color/appearance, smell, taste, texture, and overall liking); (2) a Check-All-That-Apply (CATA) test [20]; (3) an indication of the perceived intensity of selected descriptors; and (4) a short questionnaire on socio-demographics and habits regarding consumption and purchasing of Parmigiano Reggiano cheese.

Firstly, consumers were asked to rate their liking of each cheese on a 9-point labeled category scale (1 = “dislike extremely”; 9 = “like extremely”). For each sample, five hedonic questions were presented, i.e., “*Please indicate how much you like this Parmigiano Reggiano in terms of color/smell/taste/texture and overall liking*”. To correctly evaluate each cheese’s texture, participants were asked to consider both its texture to the touch and in the mouth while chewing.

Secondly, the participants completed a CATA test followed by specific questions aimed at gathering information in relation to the intensity of specific attributes. In particular, this task followed an initial visual evaluation of the sample, during which the consumer was asked to select its visual attributes from a list of 4 visual descriptors with specific indications to help participants in their evaluation, specifically: “*yellow color of the cheese paste*”, “*uniformity of the cheese paste (visual characteristic evaluated on a recently cut surface which measures the degree of color inhomogeneity and therefore the presence of areas of different color compared to the predominant color)*”, “*presence of holes on the surface*”, and “*presence of tyrosine crystals (white dots on the surface of the cheese which represent an indication of the maturation of the cheese)*”. Next, they were asked to indicate the appropriateness of the perceived intensity of the “*yellow*” attribute on a Just-About-Right (JAR) scale from 1 (too little) to 5 (too much). Next, they were asked to fill out a CATA list of 9 olfactory attributes (i.e., “*fresh milk*”, “*white yogurt*”, “*seasoned*”, “*grass/hay*”, “*animal/stable*”, “*vegetable*”, “*rennet*”, “*olfactory pungent/irritating*”, and “*cheese crust*”) followed by giving an indication of the appropriateness of the perceived intensity for the “*fresh milk*” attribute. Consumers were also asked to answer a CATA list of 7 taste/mouthfeel descriptors (i.e., “*sweet*”, “*salty*”, “*sour*”, “*bitter*”, “*umami*”, “*pungent*”, and “*astringent*”), followed by the JAR indication for “*sweet*” and “*salty*” tastes. They were then asked to answer a CATA list of 4 texture descriptors, each defined with a brief explanation to allow consumers to correctly evaluate the samples, specifically: “*hardness (mechanical characteristic that measures the resistance offered by the sample to a slight pressure exerted by the molars before deformation or breakage)*”, “*friability (the ability of the sample, at the beginning of chewing, to generate numerous fragments)*”, “*graininess (perception of feeling the rounded grains in the cheese, more or less fine, during and at the end of chewing)*”, and “*solubility (sensation measures how quickly chewed cheese dissolves in saliva)*”, followed by the JAR indication for “*friability*” and “*graininess*”.

These lists of descriptors were developed considering those reported in the PDO (Protected Designation of Origin) specification sheet and previous literature [21,22,23,24,25], and from 5 informal elicitation sessions (i.e., 5 h) performed with 10 people from the staff of the Department of Agricultural and Food Sciences (Campus of Cesena, Italy). The attributes were presented in a balanced order, and for each product and each subject, the sequence of attributes in the list was randomized. Each consumer had to select all the attributes they perceived in the sample. Between the products, a rinsing with water of the mouth was performed.

Finally, the questionnaire aimed to collect information about each respondent’s age, gender, and nationality, the frequency of their consumption of Parmigiano Reggiano cheese (every day, 4–5 times per week, 2–3 times per week, once a week, every two weeks, once a month, or “I do not consume Parmigiano Reggiano cheese”), what information on the label they consider to be important when choosing Parmigiano Reggiano (i.e., seasoning, origin of the milk, organic, mountain product, direct knowledge of the dairy, and minimum durability date), and the seasoning and type of Parmigiano Reggiano product they usually buy (i.e., 12–19 months, 20–26 months, 27–34 months, 35–45 months, 46–79 months, or more than 80 months seasoned; discounted product, grated product, portioned product (snack), or other format: “*please specify*”).

The subjects were asked not to eat, drink, smoke, or wear perfume or lipstick for 1 h before the evaluation session. The participants took approximately 30 min to complete the test.

The test was conducted in a lecture room, during breaks, so that the Parmigiano Reggiano could be consumed as a snack. Data were collected with Qualtrics^®^ online survey software (Qualtrics^®^, LLC; Seattle, WA, USA) and individuals directly answered test questions using their mobile phone.

### 2.5. Image Analysis

Additionally, the samples were subjected to image analysis to assess their color differences using an instrumental technique in addition to the consumer sensory evaluation. Image analysis was carried out using an “electronic eye” (Visual Analyzer VA400 IRIS, Alpha MOS, Toulouse, France), a high-resolution (2592 × 1944 p) charge-coupled camera equipped with a photo camera (16 million colors). This instrument is furnished with two lights (2 × 2 fluorescent tubes) with a color temperature of 6700 °K; only the light that shines from above was used to take pictures of the samples. Samples were placed on a white plastic tray, diffusing a uniform light inside the device’s closable light chamber, and the CCD camera took a picture. The instrument was calibrated with a certified color checker (ColorChecker Cclassic, x-Rite, Grand Rapids, MI, USA) before taking the pictures.

### 2.6. Data Analysis

All data analyses were performed using XLStat 2023 1.1 (Lumivero, Addinsoft, Boston, MA, USA). CATA data were elaborated by performing a calculation of the number of times each descriptor was chosen by consumers for each sample to obtain four occurrence matrices, one for CATA results related to visual attributes, one for olfactory descriptors, one for taste and mouthfeel sensations, and the last for texture descriptors. To test the significance of these attributes in discriminating between the samples, Cochran’s Q tests were performed on the occurrence matrices. In the end, a penalty analysis (PA) was performed between the significant attributes and the overall liking to find the positive and negative drivers of liking for the samples. The PA included graphical representations of the CATA data. These graphs displayed the percentage of participants selecting a specific attribute across the samples on the X-axis, while the Y-axis represented the average impact of these attributes on overall liking. The JAR results were combined with the overall liking ratings and analyzed using a PA to estimate the impact of each attribute being “*much too weak*” or “*much too strong*” on the overall liking. Student’s t-tests (*p* ≤ 0.05) were performed on the liking ratings. Finally, the software Alphasoft version 14.0 (Alpha MOS, Toulouse, France) was used to explore the data from the image analysis using a principal component analysis (PCA).

## 3. Results

### 3.1. Socio-Demographic Characteristics of Participants 

The individuals were mostly from the Emilia-Romagna region, in particular from the provinces of Forlì-Cesena (54 subjects, 45%), Ravenna (17 subjects, 14%), Bologna (10 subjects, 8%), Rimini (10 subjects, 8%), Pesaro–Urbino (3 subjects, 3%), Treviso (3 subjects, 3%), Bari (2 subjects, 2%), Chieti (2 subjects, 2%), Como (2 subjects, 2%), Macerata (2 subjects, 2%), Palermo (2 subjects, 2%), Ancona (1 subject, 1%), Aquila (1 subject, 1%), Ascoli-Piceno (1 subject, 1%), Bergamo (1 subject, 1%), Firenze (1 subject, 1%), Frosinone (1 subject, 1%), Modena (1 subject, 1%), Repubblica di San Marino (1 subject, 1%), Roma (1 subject, 1%), Sondrio (1 subject, 1%), Teramo (1 subject, 1%), and Venezia (1 subject, 1%).

Table 1 shows the frequency of elicitation of the items related to several claims that could be reported on the label and the type of Parmigiano Reggiano PDO in terms of seasoning and the type of product. The most important claim for respondents is the seasoning, while the most selected product was Parmigiano Reggiano seasoned for 20–26 months. As reported in Table 1, the organic indication was not considered by most subjects at the time of purchase; also, most subjects do not buy portioned Parmigiano Reggiano PDO, and no one selected the seasoning of 46–79 months.

The majority of those interviewed were frequent consumers of Parmigiano Reggiano cheese, and in fact, 86 respondents declared that they eat it every day or between 2 and 4 times a week.

On the other hand, five individuals declared that they do not eat Parmigiano Reggiano, although they decided to test the samples and answer the questionnaire. Since no information was acquired in relation to their consumption habits of hard cheeses in general and participation in this study was voluntary, it can be assumed that they eat other types of cheese.

### 3.2. Hedonic Assessment of Parmigiano Reggiano Samples

For the two samples tested (P-DG and P-FG), no significant differences were highlighted in terms of liking (appearance, smell, taste, texture, and overall) (Figure 1). In fact, the mean scores were: 7.076 (P-DG) and 7.303 (P-FG) for visual liking, 7.168 (P-DG) and 7.328 (P-FG) for olfactory liking, 7.252 (P-DG) and 7.294 (P-FG) for liking the taste, 6.950 (P-DG) and 7.084 (P-FG) for liking the texture, and 7.059 (P-DG) and 7.303 (P-FG) for overall liking. However, considering the ratings obtained for the overall liking of the two samples, it can be assumed that subjects liked them moderately (a rating of 7 on the 9-point hedonic scales corresponds to “*like moderately*”).

### 3.3. Check-All-That-Apply

Table 2 shows the number of citations for each of the attributes of the CATA question used to describe the Parmigiano Reggiano samples and the relative *p*-value (Cochran’s Q tests). The most frequently used terms were “*yellow*” and “*presence of tyrosine crystals*” for appearance; “*seasoned*”, “*cheese crust*”, and “*fresh milk*” for odor; “*salty*”, “*sweet*”, and “*umami*” for taste; and “*graininess*” for texture.

Significant differences (*p* < 0.05) were found in the frequencies of three out of four terms of the CATA question related to appearance, four of the nine odor terms; two of the seven taste/mouthfeel descriptors, and for one of the texture attributes (Table 2).

According to the Cochran’s Q test results, three visual attributes, five smell descriptors, two taste attributes, and one texture descriptor of the Parmigiano Reggiano PDO cheese tested were significant to discriminate between the samples (Table 2). Furthermore, the taste attribute “*bitter*” and the texture attributes “*friability*” and “*graininess*” showed *p*-values very close to significance (α = 0.05) (Table 2). Thus, a total of 14 descriptors (3 related to appearance, 5 to smell, 3 to taste, and 3 to texture) were included in the principal coordinate analysis (PCoA), together with the overall liking scores (Figure 2). As shown in Figure 2, it was highlighted that liking is negatively related to “*yellow*”, even if the negative correlation is very weak (−0.072). Moreover, the attributes “*uniformity*” and “*presence of tyrosine crystals*” were strongly and negatively correlated (−0.538), indicating that when subjects chose “*uniformity*”, they did not tick “*presence of tyrosine crystals*”, and vice versa. Liking was also negatively related to “*cheese crust*” (−0.084) and “*sour*” (−0.098). Strong correlations were found for the smell attributes “*seasoned*” and “*fresh milk*” (−0.652), “*pungent*” and “*fresh milk*” (−0.534”), and “*white yoghurt*” and “*fresh milk*” (0.564).

### 3.4. Attribute Adequacy: Just-About-Right (JAR) Scale and Penalty Analysis to Assess the Relation with Liking

The results obtained for the JAR scales are reported in Figure 3. In both samples of Parmigiano Reggiano PDO, most of the parameters were scored in the “JAR” category, ranging from 45% (“*fresh milk*” and “*friability*”) to 68% (“*salty*”) for the P-DG sample and from 50% (“*yellow*”) to 65% (“*salty*”) for P-FG. The attribute “*fresh milk*” for sample P-FG was ranked as “too low” by 52% of respondents. Another interesting result, also related to sample P-FG, is related to the “*yellow*” descriptor: in particular, 49% of individuals reported that the color of this sample was “too much” (Figure 3).

A penalty analysis was carried out in order to understand which of the selected attributes affected the acceptability of the Parmigiano Reggiano PDO samples to a greater or a lesser extent. In fact, penalization indicates how much the overall liking of a specific product decreases when a specific attribute is considered “too much” or “too low”; thus, the higher the obtained penalty values, the greater the impact on acceptability [26]. Table 3 shows the results of the penalty analysis for the two Parmigiano Reggiano PDO samples. In particular, this table reports the six variables (i.e., attributes), the three levels (i.e., “not enough”, “JAR”, and “too much”), the percentages of answers for each level and each attribute, the mean drops, the penalties, and the related *p*-values. If the number of responses was below the threshold of 20%, the penalizations were not considered, in line with the study by Ortega-Heras et al. (2019) [26]. 

Regarding sample P-DG (i.e., the Parmigiano Reggiano produced with milk from cows fed dry hay), several attributes were considered as “not enough”, which affected the overall liking of this sample. In particular, this product was highly penalized for not being yellow enough, as well as for its saltiness, graininess, and friability. The greater penalization in terms of liking was identified for the attribute “*graininess*” (penalties = 1.232) (Table 3). On the other hand, for the attributes “*fresh milk*” and “*sweet*”, the mean decreases were not statistically significant, while the penalties were (*p*-values 0.025 and 0.002, respectively), indicating that these attributes matter for consumers, even if our survey may not have had sufficient power to detect which specific mean drop (not enough and/or too) was responsible.

Concerning the P-FG sample (Parmigiano Reggiano from cows fed fresh forage), only two attributes affected the overall liking, i.e., “*yellow*” and “*sweet*”. In this case, “too yellow” determined a penalty of 0.667, while “not sweet enough” had a penalty of 0.613. “*Graininess*” and “*friability*” were considered important attributes for respondents (*p*-values of the penalties 0.014 and 0.008, respectively), but the survey was not strong enough to detected which specific mean drop was responsible for this (Table 3).

It is well reported that gender has a significant impact on sensory responses and food preferences [27]. Thus, to assess if the adequacy of the specific attributes that differently affect liking for females and males, the JAR and liking data were elaborated according to gender (Table 4). As shown in Table 4, for women, low intensities of “*fresh milk*” flavor, “*sweet*” taste, and “*grainy*” texture determined a penalization for the Parmigiano Reggiano PDO sample from dairy cows fed dry hay (P-DG). The penalizations were: 0.960 for a perceived intensity of “*fresh milk*” considered as “*not enough*”, 1.081 if the sweetness was considered “*not enough*”, and 1.493 if the graininess was considered “*not enough*”. On the other hand, for men, the “*not sweet enough*” observation had an impact on their liking, and in particular, the penalization was 0.613, which is less in comparison with the results for the women. Moreover, for the same samples, among the men, the perception of “*too yellow*” also penalize the product (penalization of 0.667) (Table 4).

Regarding the results related to the P-FG sample (Parmigiano Reggiano PDO from dairy cows fed fresh forage), as reported in Table 4, the perception of “*too yellow*” penalized (0.782) the product for women, while for men, it had no significant impact on liking.

### 3.5. Image Analysis 

The software used with the photo camera (Alphasoft, version 14.0, Alpha MOS, Toulouse, France) allowed the color spectra to be grouped into categories of 16 bits for each RGB coordinate, resulting in 4096 variables that were analyzed. The proportion of each color in the image, on a fixed scale of 4096 colors, is represented as a percentage. 

## 4. Discussion

This study investigated consumer perception and acceptability of Parmigiano Reggiano PDO cheeses, seasoned for 24 months, during a sensory test in a pseudo-natural food situation. The subjects, in fact, tasted the samples in a lecture room during their mid-morning break as if the Parmigiano Reggiano samples were a snack. The choice to use the term “pseudo-natural” was made in relation to what has been previously reported by Torri and Salini (2016) [25]. In their investigation, a sensory test was performed in the context of an agri-food sector fair, simulating the free multiple-tasting experience that a visitor could experience in a food exhibition [25]; in our case, we wanted to simulate the experience of a mid-morning snack that students often have. Thus, the test conducted herein was carried out in an environment where students usually consume food, even if the tested consumption was not wholly natural. In fact, no timing constraints were imposed, but social interactions were not allowed, and the participants were required to follow several instructions. In particular, they were asked to apply a specific evaluation procedure and to taste a precise number of products following a specified sequence. For these reasons, the pseudo-natural situation is considered as a distinct testing condition. This differs from a natural setting characterized by spontaneous consumption, as well as from the imitation of a natural setting or a central location condition with a controlled setting [25,28,29]. The conditions applied for this study allowed the participants to describe the sensory properties of Parmigiano Reggiano PDO cheeses, and they found significant differences among the samples for several attributes. It is essential to underline that since the consumers tested two samples that differed in terms of color (Table 5), a significant difference in relation to perceived color was expected. In particular, the samples were both seasoned for 24 months and produced from the same dairy. The color difference between the two samples was related to the feed type: one sample was manufactured from milk belonging to dairy cows fed dry hay, while the second was obtained from animals fed fresh forage. As reported in Table 5, sample P-FG was characterized by color with higher b* (i.e., a higher yellow color) and lightness values. 

The two samples were significantly different when described by consumers. It is necessary to highlight that consumers did not receive a list of attribute definitions for all attributes. In particular, definitions were reported only in the case of the texture attributes, for which they were considered important for the comprehension of the descriptors “*friability*” and “*graininess*”, and for two of the four visual descriptors (i.e., “*uniformity of the cheese paste*” and “*presence of tyrosine crystals*”). Thus, in relation to the CATA terms for appearance, smell, and taste, it should be considered that respondents might use and interpret these CATA terms differently from a trained panel. In fact, from a methodological point of view, this study adopted the “Check-All-That-Apply” method as a rapid technique to obtain sensory descriptions of Parmigiano Reggiano PDO samples from consumers. CATA tests have been previously applied to investigate the sensory profile and characteristics of different cheeses [30,31,32,33].

In the current study, consumers described the Parmigiano Reggiano manufactured from milk obtained from dairy cows fed dry hay as having attributes typical of mildly seasoned Parmigiano Reggiano cheese, i.e., “*fresh milk*” and “*solubility*”. In contrast, the cheese from dairy cows fed fresh forage was described as having attributes that are generally more frequently perceived in more-seasoned Parmigiano Reggiano PDO, such as “*yellow*”, “*seasoned*”, “*pungent*”, and “*cheese crust*” [34]. These results are interesting considering that they were perceived differently even if the two samples had the same seasoning time. This could be related to differences in the type of feed; in fact, it is well known that different feeds influence the sensory properties of milk and cheese. In particular, the highest intensity of yellowness perceived in sample P-FG could be probably related to a higher content of β-carotene in the P-FG milk [17]. However, even if several attributes were significant in discriminating among the samples, they did not seem to be directly related to the overall liking (Figure 2). In fact, generally, penalty analyses of the CATA frequencies of all the attributes were used in order to assess the drivers of liking and disliking scores [25]. Some of the attributes that were significant in discriminating between the products, i.e., “*yellow*”, “*cheese crust*”, “*bitter*”, and “*friability*”, clearly had a negative correlation with the hedonic judgment (Figure 2). Moreover, as was also found by Torri and Salini (2016) [25], “*umami*” was not significant in terms of product discrimination; as also previously highlighted by the abovementioned authors, this result may be attributable to the Italian consumers experiencing difficulty in recognizing the “*umami*” attribute. However, as previously reported, it should be emphasized that the CATA method does not provide direct information on either the intensity of the attribute or on whether an attribute is liked in relation to the intensity with which it exists in the product [25]. Thus, to more thoroughly investigate this aspect, we also investigated the adequacy of the intensity of specific attributes. It was decided not to evaluate all of the selected attributes for each assessor to avoid the questionnaire becoming too long. In fact, in this case, if an assessor had selected many attributes during the CATA, they would have had to indicate the adequacy of each one on the JAR scale. Instead, some attributes were pre-selected, as previously described (Section 2.4), and rated by all assessors in terms of the adequacy of their intensity. The JAR scale indicated that the Parmigiano Reggiano PDO sample from cows fed fresh forage was perceived by 49 of the participants as “*too yellow*” (Figure 3). According to Manzocchi et al. (2021) [17], cheeses produced from milk obtained from dairy cows fed fresh forage are perceived as yellower than cheeses from silage- and hay-fed cows. In addition, previous studies have reported a lower content of carotenoids in milk obtained from silage-fed cows, in particular from hay-fed cows [35,36], thus influencing its sensory properties and, in particular, leading to a more intense yellow hue [36]. These differences in color could impact consumer acceptance, with some associating the yellow color of dairy products with naturally fed cows [12]. The results shown in Figure 3 are in line with what has been reported by Manzocchi et al. (2021) [17]; in fact, the majority of participants perceived the Parmigiano Reggiano PDO sample from cows fed dry hay as “*yellow JAR*” or “*low yellow*”. In addition, as previously mentioned, consumers perceived the “*fresh milk*” flavor differently in the two samples. In fact, sample P-FG was rated by the majority of the participants as “*low fresh milk*” (Figure 3), which is in line with what was reported by Manzocchi et al. (2021) [17] in relation to lactic odor. Moreover, even if neither the “*graininess*” nor “*friability*” were statistically significant in discriminating among samples (Table 2), a greater number of individuals indicated the intensity of these two attributes as being “*low*” in the sample of Parmigiano Reggiano from cows fed dry hay compared to the sample of cheese from cows fed fresh forage. This result is in line with what some authors have previously reported, according to which cheeses produced from ‘pasture milk’ were rated higher in terms of friability and graininess. In that case, the authors directly attributed this to the variation in the fatty acid profiles of the milks [12].

To investigate the relationship between the attributes and liking, penalty analyses were performed on the JAR and liking rates. JAR scaling measures each attribute’s performance using the concept of a consumer’s ‘ideal’, and thus it is assumed that the participants have an implicit ideal point in their mind [37]. Our results showed that the highest negative impact on liking, equal to a penalty of 1.232 for the sample of Parmigiano Reggiano PDO from cows fed dry hay, was related to a *graininess* that was rated as “*not enough*” (i.e., *too low*) (Table 3). This underlines the importance of the presence of graininess, which is considered one of the typical attributes of Parmigiano Reggiano PDO [22]. Moreover, yellowness had an impact on liking. In particular, a high penalty (0.961) was found for the adequacy of the yellow color in sample P-DG when the attribute was evaluated as “*not enough*” (Table 3). On the other hand, for sample P-FG, the perception of it being “*too yellow*” determined a lower penalty on the liking, equal to 0.667. However, no statistically significant differences were found in terms of liking between the two samples (Figure 1).

Data related to the JAR scale and liking were also elaborated according to gender in order to highlight possible gender-related differences (Table 4). Such differences were noted for sample P-DG (Parmigiano Reggiano from cows fed dry hay). In fact, as reported in Table 4, for women, several attributes had substantial impact on their overall liking of this sample. In particular, the adequacy of the graininess being rated as “*not enough*” led to a penalty equal to 1.493 for women, while no significant penalty was highlighted for this attribute for male participants (Table 4). In addition, for women, the “*sweet*” and “*fresh milk*” attributes being rated as “*not enough*” led to significant penalties of 0.960 and 1.081, respectively. Whereas for males, significant penalties were highlighted for the “*yellow*” attribute when rated as “*too*” and the “*sweet*” attribute when rated as “*not enough*”; however, the impact on liking was lower (around 0.6) in both cases (Table 4).

In relation to the claims made on the label, the most important was seasoning, which was selected by 99 participants (Table 1). This result is interesting, especially in relation to the fact that the majority of these consumers also declared that they usually buy Parmigiano Reggiano PDO seasoned between 16 and 20 months (Table 1). Recently, investigations on Parmigiano Reggiano PDO cheese aged for 24 months have increased [1,5,25], but to date, there has been little research focus on consumer liking, perception, and consumption habits of Parmigiano Reggiano PDO aged for 24 months. One hundred of the participants in the present study declared that they consume Parmigiano Reggiano from once a week to every day, which is in line with the well-known fact that this product is common in the diet of the Italian population, and is also a part of the Mediterranean diet [38].

## 5. Conclusions

This sensory evaluation of these two products revealed that they had distinct sensory profiles. The Parmigiano Reggiano produced with milk from cows fed with fresh forage had a higher intensity of yellowness, as emerged both from the image analysis and sensory evaluation of the yellow attribute, and it was characterized by attributes generally related to a more-seasoned Parmigiano Reggiano, e.g., “*yellow*”, “*seasoned*”, “*cheese crust*”, and “*pungent*”. On the other hand, the cheese from cows fed with dry hay was described with attributes typically related to a less-seasoned Parmigiano Reggiano PDO cheese, such as “*fresh milk*”. Both samples were highly liked, which was potentially due to the high quality of the product associated with the production technology, regulated by the PDO specification sheet. Furthermore, this study has highlighted the concept that the consumers interviewed had specific expectations for certain attributes of this product. In particular, they have expectations, somehow a preconceived idea, about the intensity of the yellow color and granularity, which could serve as significant drivers in the domestic market and also as crucial factors to be locally declined for the international market. For example, the more intense yellow color of the Parmigiano Reggiano from cows fed with fresh forage could be a driver of preference for this product in markets where consumers appreciate cheeses with richer yellow hues. This can be relevant to establish which product is most suitable for exportation to different areas of the world. Furthermore, understanding these sensory attributes is crucial to properly disseminate information about the product, especially among consumers who may be less familiar with it.

## Figures and Tables

**Figure 1 foods-13-00309-f001:**
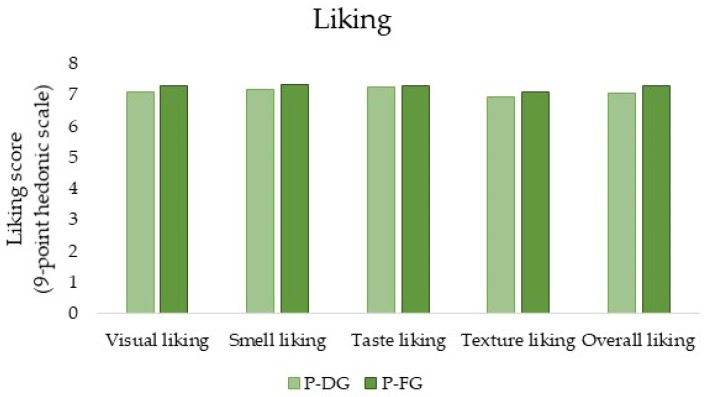
Histogram depicting the results of liking scores in terms of appearance, smell, taste, texture, and overall liking for the two samples (P-DG = Parmigiano Reggiano produced with milk from cows fed dry hay; P-FG = Parmigiano Reggiano produced with milk from cows fed fresh forage) (Student’s *t*-test, *p* ≤ 0.05, Tukey’s HSD).

**Figure 2 foods-13-00309-f002:**
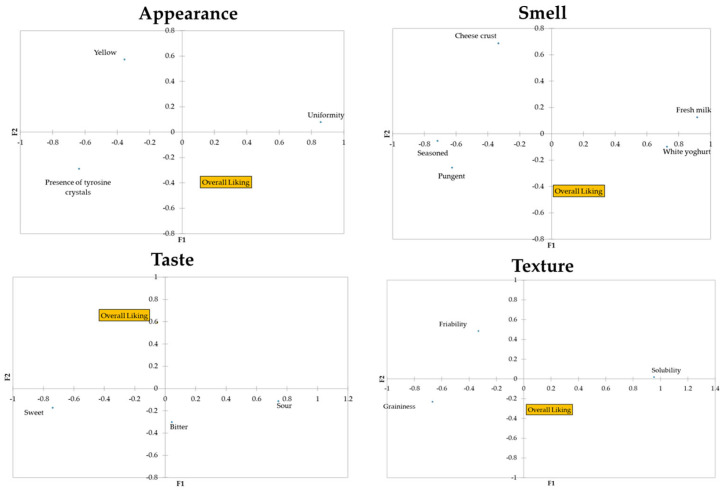
Results of the principal coordinate analysis (PCoA) of the statistically significant CATA descriptive attributes (Cochran’s Q test).

**Figure 3 foods-13-00309-f003:**
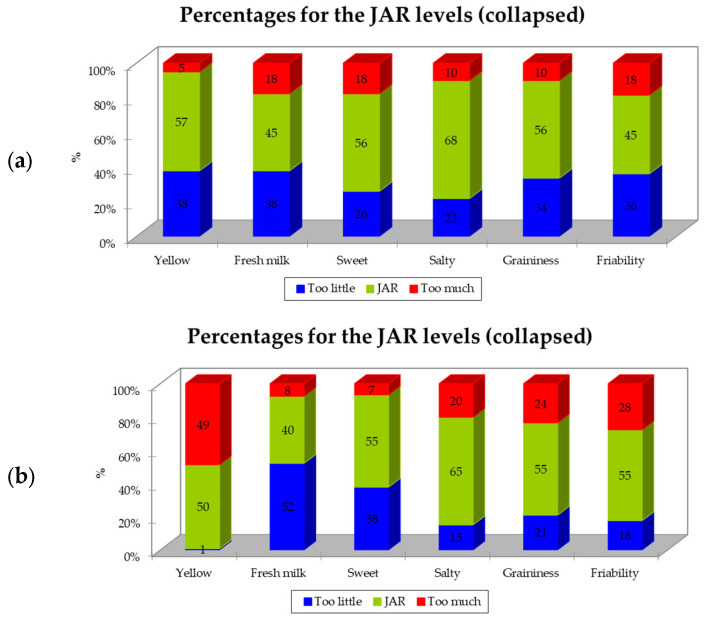
Just-About-Right (JAR) scale percentages of responses grouped in three levels for samples of Parmigiano Reggiano PDO. (**a**) P-DG—Parmigiano Reggiano produced with milk from cows fed dry hay; (**b**) P-FG—Parmigiano Reggiano produced with milk from cows fed fresh forage.

**Table 1 foods-13-00309-t001:** Frequency of elicitation of the information on the label that the interviewed subjects (*n* = 119) considered important when choosing Parmigiano Reggiano, and the seasoning and type of Parmigiano Reggiano product that they usually buy.

Claim on the Label	Frequency	Type of Parmigiano Reggiano PDO	Frequency
Seasoning	99	Discounted product	34
Best-before date	30	Grated product	21
Organic	14	Portioned product (snack)	7
Origin of the milk	37	12–19 months	42
Direct knowledge of the dairy factory	24	20–26 months	60
Mountain product	17	27–34 months	31
		35–45 months	13
		46–79 months	0

**Table 2 foods-13-00309-t002:** Results of the Check-All-That-Apply question: Number of citations for each attribute and sample and the relative *p*-values (Cochran’s Q tests) (P-DG = Parmigiano Reggiano produced with milk from cows fed dry hay; P-FG = Parmigiano Reggiano produced with milk from cows fed fresh forage).

	Attribute	*p*-Value	Sample	
			P-DG	P-FG
Appearance	Yellow	**<0.0001**	49	103
	Uniformity	**<0.0001**	87	52
	Presence of holes	0.336	13	18
	Presence of tyrosine crystals	**0.005**	53	75
Smell	Fresh milk	**<0.0001**	56	24
	White yogurt	**0.012**	20	9
	Seasoned	**<0.0001**	48	93
	Grass/hay	0.117	31	22
	Animal/stable	0.237	21	28
	Vegetable	0.285	11	7
	Rennet	0.732	29	27
	Pungent	**<0.0001**	0	38
	Cheese crust	**0.013**	57	74
Taste/mouthfeel	Sweet	**0.000**	57	31
	Salty	0.465	80	90
	Sour	**<0.0001**	38	66
	Bitter	0.059	19	11
	Umami	0.117	59	55
	Pungent	0.513	22	34
	Astringent	0.862	28	30
Texture	Hardness	0.105	25	35
	Friability	0.093	65	76
	Graininess	0.091	67	80
	Solubility	**0.002**	68	44

**Table 3 foods-13-00309-t003:** Percentages, mean drops, penalties, and related *p*-values of the “too little” or “too much” responses on the 5-point JAR scale for “*yellow*”, “*fresh milk*”, “*sweet*”, “*salty*”, “*graininess*”, and “*friability*” for the two samples of Parmigiano Reggiano PDO (P-DG = Parmigiano Reggiano produced with milk from cows fed dry hay; P-FG = Parmigiano Reggiano produced with milk from cows fed fresh forage).

Variable	Level	P-DG						
		%	Sum (Overall Liking)	Mean (Overall Liking)	Mean Drops	*p*-Value	Penalty	*p*-Value
	Not yellow enough	37.82%	296	6.578	0.893	**0.001**		
Yellow	JAR	57.14%	508	7.471			0.961	**0.000**
	Too yellow	5.04%	36	6.000	1.471			
	Milk not fresh enough	37.82%	309	6.867	0.511	0.073		
Fresh milk	JAR	44.54%	391	7.377			0.574	**0.025**
	Too fresh milk	17.65%	140	6.667	0.711			
	Not sweet enough	26.05%	215	6.935	0.468	0.067		
Sweetness	JAR	56.30%	496	7.403			0.788	**0.002**
	Too sweet	17.65%	129	6.143	1.260			
	Not salty enough	21.85%	161	6.192	1.153	**0.000**		
Saltiness	JAR	68.07%	595	7.346			0.898	**0.001**
	Too salty	10.08%	84	7.000	0.346			
	Not grainy enough	33.61%	250	6.250	1.347	**<0.0001**		
Graininess	JAR	56.30%	509	7.597			1.232	**<0.0001**
	Too grainy	10.08%	81	6.750	0.847			
	Not friable enough	36.13%	282	6.558	0.775	**0.010**		
Friability	JAR	45.38%	396	7.333			0.503	**0.049**
	Too friable	18.49%	162	7.364	−0.030			
Variable	Level	P-FG						
		%	Sum (Overall Liking)	Mean (Overall Liking)	Mean Drops	*p*-Value	Penalty	*p*-Value
	Not yellow enough	8	8.000	−0.367				8
Yellow	JAR	458	7.633			0.667	**0.001**	458
	Too yellow	403	6.948	0.685	**0.001**			403
	Milk not fresh enough	456	7.355	−0.001	0.998			456
Freshness of the milk	JAR	353	7.354			0.087	0.686	353
	Too fresh milk	60	6.667	0.688				60
	Not sweet enough	318	7.067	0.509	**0.018**			318
Sweetness	JAR	500	7.576			0.613	**0.003**	500
	Too sweet	51	6.375	1.201				51
	Not salty enough	120	6.667	0.710				120
Saltiness	JAR	568	7.377			0.210	0.339	568
	Too salty	181	7.542	−0.165	0.533			181
	Not grainy enough	175	7.000	0.530	0.113			175
Graininess	JAR	497	7.530			0.511	**0.014**	497
	Too grainy	197	7.036	0.495	0.127			197
	Not friable enough	138	6.571	0.982				138
Friability	JAR	491	7.554			0.554	**0.008**	491
	Too friable	240	7.273	0.281	0.169			240

**Table 4 foods-13-00309-t004:** Percentages, mean drops, penalties, and related *p*-values of the “too little” or “too much” responses on the five-point JAR scale for “yellow”, “fresh milk”, “sweet”, “salty”, “graininess”, and “friability” for the two samples of Parmigiano Reggiano PDO (P-DG = Parmigiano Reggiano produced with milk from cows fed dry hay; P-FG = Parmigiano Reggiano produced with milk from cows fed fresh forage), according to gender of the participants (F = females, M = males).

Variable	Level	P-DG W						
		%	Sum (Overall Liking)	Mean (Overall Liking)	Mean Drops	*p*-Value	Penalty	*p*-Value
	Not yellow enough	36.76%	168	6.720	0.705	0.054		
Yellow	JAR	58.82%	297	7.425			0.782	**0.028**
	Too yellow	4.41%	18	6.000	1.425			
	Milk not fresh enough	32.35%	143	6.500	1.041	**0.010**		
Fresh milk	JAR	54.41%	279	7.541			0.960	**0.006**
	Too fresh milk	13.24%	61	6.778	0.763			
	Not sweet enough	26.47%	120	6.667	0.897	**0.012**		
Sweetness	JAR	57.35%	295	7.564			1.081	**0.002**
	Too sweet	16.18%	68	6.182	1.382			
	Not salty enough	19.12%	80	6.154	1.151			
Saltiness	JAR	67.65%	336	7.304			0.623	0.100
	Too salty	13.24%	67	7.444	−0.140			
	Not grainy enough	36.76%	159	6.360	1.446	**<0.0001**		
Graininess	JAR	52.94%	281	7.806			1.493	**<0.0001**
	Too grainy	10.29%	43	6.143	1.663			
	Not friable enough	39.71%	180	6.667	0.753	0.061		
Friability	JAR	45.59%	230	7.419			0.582	0.102
	Too friable	14.71%	73	7.300	0.119	0.054		
Variable	Level	P-DG M						
		%	Sum (Overall Liking)	Mean (Overall Liking)	Mean Drops	*p*-Value	Penalty	*p*-Value
	Not yellow enough	37.50%	113	6.278	1.241			
Yellow	JAR	56.25%	203	7.519			0.667	**0.001**
	Too yellow	6.25%	18	6.000	1.519	**0.001**		
	Milk not fresh enough	43.75%	151	7.190	−0.257	0.998		
Fresh milk	JAR	31.25%	104	6.933			0.087	0.686
	Too fresh milk	25.00%	79	6.583	0.350			
	Not sweet enough	27.08%	95	7.308	−0.188	**0.018**		
Sweetness	JAR	52.08%	178	7.120			0.613	**0.003**
	Too sweet	20.83%	61	6.100	1.020			
	Not salty enough	27.08%	81	6.231	1.144			
Saltiness	JAR	66.67%	236	7.375			0.210	0.339
	Too salty	6.25%	17	5.667	1.708	0.533		
	Not grainy enough	29.17%	83	5.929	1.416	0.113		
Graininess	JAR	60.42%	213	7.345			0.511	**0.014**
	Too grainy	10.42%	38	7.600	−0.255	0.127		
	Not friable enough	31.25%	94	6.267	0.951			
Friability	JAR	47.92%	166	7.217			0.554	**0.008**
	Too friable	20.83%	74	7.400	−0.183	0.169		
Variable	Level	P-FG W						
		%	Sum (Overall Liking)	Mean (Overall Liking)	Mean Drops	*p*-Value	Penalty	*p*-Value
	Not yellow enough	1.47%	8.000	8.000	−0.355			
Yellow	JAR	45.59%	237.000	7.645			0.753	**0.009**
	Too yellow	52.94%	247.000	6.861	0.784	**0.007**		
	Milk not fresh enough	58.82%	290.000	7.250	0.054	0.865		
Fresh milk	JAR	33.82%	168.000	7.304			0.104	0.737
	Too fresh milk	7.35%	34.000	6.800	0.504			
	Not sweet enough	41.18%	196.000	7.000	0.514	0.090		
Sweetness	JAR	51.47%	263.000	7.514			0.575	**0.047**
	Too sweet	7.35%	33.000	6.600	0.914			
	Not salty enough	14.71%	70.000	7.000	0.233			
Saltiness	JAR	63.24%	311.000	7.233			−0.007	0.981
	Too salty	22.06%	111.000	7.400	−0.167	0.653		
	Not grainy enough	19.12%	95.000	7.308	0.142			
Graininess	JAR	58.82%	298.000	7.450			0.521	0.077
	Too grainy	22.06%	99.000	6.600	0.850	**0.013**		
	Not friable enough	20.59%	96.000	6.857	0.514	0.370		
Friability	JAR	51.47%	258.000	7.371			0.281	0.338
	Too friable	27.94%	138.000	7.263	0.108	0.946		
Variable	Level	P-FG M						
		%	Sum (Overall Liking)	Mean (Overall Liking)	Mean Drops	*p*-Value	Penalty	*p*-Value
	Not yellow enough	0.00%						
Yellow	JAR	56.25%	206.000	7.630				
	Too yellow	43.75%	148.000	7.048	0.582	0.064		
	Milk not fresh enough	45.83%	166.000	7.545	−0.182	0.578		
Freshness of the milk	JAR	45.83%	162.000	7.364			−0.021	0.948
	Too fresh milk	8.33%	26.000	6.500	0.864			
	Not sweet enough	33.33%	115.000	7.188	0.433	0.182		
Sweetness	JAR	60.42%	221.000	7.621			0.621	0.051
	Too sweet	6.25%	18.000	6.000	1.621			
	Not salty enough	16.67%	50.000	6.250	1.281			
Saltiness	JAR	66.67%	241.000	7.531			0.469	0.160
	Too salty	16.67%	63.000	7.875	−0.344			
	Not grainy enough	25.00%	80.000	6.667	0.958	**0.029**		
Graininess	JAR	50.00%	183.000	7.625			0.500	0.111
	Too grainy	25.00%	91.000	7.583	0.042	0.993		
	Not friable enough	14.58%	42.000	6.000	1.786			
Friability	JAR	58.33%	218.000	7.786			0.986	**0.001**
	Too friable	27.08%	94.000	7.231	0.555	0.071		

**Table 5 foods-13-00309-t005:** Results of the image analysis performed with the “electronic eye” camera. This table reports the color code registered by the instrument and the relative CIELab values for each color found for the two samples of Parmigiano Reggiano PDO (P-DG = Parmigiano Reggiano produced with milk from cows fed dry hay; P-FG = Parmigiano Reggiano produced with milk from cows fed fresh forage).

Color (Electronic Eye)	P-DG			P-FG		
	L *	a *	b *	L *	a *	b *
2694	59.160	9.238	22.064			
2949	60.699	14.775	33.176			
2964	64.617	5.474	45.817			
2965	64.776	6.409	38.084			
2966	64.973	7.557	29.947			
2967	65.210	8.922	21.596			
2983	69.380	0.699	27.074			
3222	66.672	14.239	32.510			
3238				70.706	6.023	37.490
3239				70.913	7.238	29.411
3254				74.858	−2.006	42.494
3255				75.046	−0.816	34.668
3256				75.268	0.568	26.636
3590				76.363	4.605	44.713
3511				76.545	5.694	36.925
3512				76.759	6.963	28.921

## Data Availability

Data are contained within the article, and openly available in AMS Acta at https://doi.org/10.6092/unibo/amsacta/7492.

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
