# Peer review of "Consumer Perception and Liking of Parmigiano Reggiano Protected Designation of Origin (PDO) Cheese Produced with Milk from Cows Fed Fresh Forage vs. Dry Hay"

_foods, 2024, doi:10.3390/foods13020309_

Round 1

Reviewer 1 Report

Comments and Suggestions for Authors

Comments to Authors

The manuscript ID foods-2809868 is related with the consumer perception and liking of Parmigiano Reggiano PDO cheese produced with milk from cows fed fresh forage. It contains useful and interesting information which could improve the knowledge and make advances in the scientific and production fields. After analysis of the manuscript, generally, I notice that the manuscript needs some changes. 

Thus, there are some suggestions to improving the manuscript:

Abstract

The abstract needs revision. 

Authors use at least 5 times the word “liking”. This word is difficult to understand in some sentences because the meaning of this word is related with “taste”. Thus, I suggest you revise the use of this word in some sentences. 

Another suggestion is changing the word “subjects” to individuals/ panellist/ tasters or similar.

Lines 25-28: Are these good or bad characteristics of the cheeses?

Lines 28-31: This sentence is not clear enough.  The authors need a simpler sentence that the reader can understand.

Keywords

You need to revise this section. For instance, if you write fresh forage, why didn´t you choose the other type of feeding the cows? Again, the word liking is present, and, in my opinion, it is not a good keyword considering this investigation. On the other hand, it would make perfect sense for o.ne of the keywords to be sensory analysis. Please provide good keywords

Introduction

Very precise, well written and organized.

Material and methods

Generally, some aspects should be provide considering the samples. For example:  How many cheeses did you have in each batch? How many samples? Are the cheeses from the same lot or from different lots? These are not clear enough.

Line 107: Provide another word for “subjects” (see a previous comment related with this one).  The same comment in line 170.

In general, it would be useful for the authors to mention which of the attributes studied are good and which are not, considering the type of cheese studied. For example, is cheese with large holes good or bad? Is the colour being more yellow a good parameter? Making these and other references would enlighten readers and make them more informed. It could be in this section or another. The same general comment for conclusions section

Results and discussion

Precise, well written and organized.

However, I suggest you change the title of point 3.1.

Due to the comments above, it is my opinion that the manuscript needs a major revision to be consider again for evaluation.

Comments on the Quality of English Language

Minor editing of English language required.

Reviewer 2 Report

Comments and Suggestions for Authors

The research evaluated the use of two different types of feed for dairy cows and the consequent consumer perceptions regarding the sensory aspects of Parmigiano Reggiano PDO cheese. The results are intriguing and contribute to a better understanding of this still underexplored subject.

Overall, the article is well-written and structured, with the introduction, Materials and Methods, results, and discussion sections being well-crafted and comprehensible. Below, I suggest some corrections to enhance the manuscript's quality.

·        Regarding the manuscript's title, it is my belief that it needs to reflect the fact that the research actually compared different feeding methods (dry hay vs. fresh forage) for the animals, rather than solely utilizing fresh forage as implied.

·        Concerning the abstract, it is well-developed, but the names of the statistical tests should be omitted, retaining only the "p-value" or mentioning "statistically significant differences." The presentation of specific statistical tests in the abstract is not customary.

·        The keywords should not contain words already included in the manuscript's title.

·        The sentence located between lines 40 and 42 is incomplete and needs to be finalized.

·        Line 73 – Remove "in general, and its minor components in particular."

·        The authors need to clarify how many repetitions were conducted in the study. Was only one batch produced for each type of cheese? Was it a one-time occurrence?

·        Do the authors have information about the herds used in milk production? Details such as breeds, lactation, and the number of animals and/or farms contributing to each batch of milk should be provided.

·        Figure 1 is redundant and can be presented as text, as it only characterizes the research participants and is not a crucial research outcome. Given the abundance of figures in the manuscript, more relevant information can be conveyed without it. Values should be presented as percentages (%) rather than raw numbers. Additionally, I recommend removing Figure 2 and presenting the information as text.

·        There are no results of statistical tests for section 3.1?

·        Line 266 – Retain only the reference to the figure without presenting the p values.

·        Regarding Tables 3 and 4, the authors may reconsider presenting them differently, perhaps dividing them into two tables. In the current format, their presentation may be compromised in the edited version.

·        The conclusion needs to be reformulated. It should be precise and direct in addressing the proposed objective. What has been concluded? It should not merely reiterate the results.

Reviewer 3 Report

Comments and Suggestions for Authors

Consumer perception and liking of Parmigiano Reggiano PDO cheese produced with milk from cows fed fresh forage

Manuscript ID foods-2809868

Comments;

1-    Did the authors consider the parity factors for cow, how the authors select cows for feeding and batch preparation?

2-    The novelty of study should be emphasized

3-    The authors should explain the differences among the results at the same time

4-    Figure 4 is not clear

5-    The impact of atherogenic index and thrombogenic index on the quality of cheese should be discuss

6-    The authors should elaborate the sensory operational conditions at the time of sensory test and inclusion and exclusion criteria for panelist

7-    How much quantity of cheese was offered for each panelist

8-    In material and methods some procedures are presented without citation.

Comments on the Quality of English Language

English language is fine only minor editing is required

Reviewer 4 Report

Comments and Suggestions for Authors

The article deals with the evaluation of the influence of feed on cheese quality. This cheese is very popular in many countries around the world, so the article can be evaluated as beneficial, as it contributes to the improvement of an important food product. The methodology according to which the research was solved is very sophisticated, the authors proceeded systematically. Nevertheless, I have several questions and comments about the submitted article.

1. keywords should not contain words from the title of the article

2. please, modify the graphs in Figures 1 – 3 according to the instructions for the authors, especially to add the names of the quantities on the vertical axes.

3. if possible, it would be interesting to supplement the text of the article with several photos from the course of the experiments.

4. the assessment was elaborated very systematically, with an effort to assess the quality as objectively as possible from the consumer's point of view.

5. on lines 93 – 94 it is stated: "The two batches were coded as P-DG (Parmigiano Reggiano from cows fed 15-18 kg of dry hay/cow/day) and P-FG (Parmigiano Reggiano from cows fed 40 kg of fresh forage/cow/day plus 9-12 kg of dry hay).” - It seems to me a certain inequality in nutrition. According to the availability or unavailability of fresh forage, can it be concluded that, based on the results of this research, the quality of the examined Parmigiano Reggiano cheese produced from milk from cows from the winter period, when fresh forage is unavailable, will be worse?

6. I wonder if the authors of the article also considered the durability of the product - cheeses, which is one of the factors that consumers are very often interested in, especially in the case of export. How, in your opinion, does the age of the cheese manifest itself, in the form of sale (slicing and packing into small sizes, etc.)?

7. I would be interested in the opinion of the authors on the issue of "grating cheese". Considering that this cheese is very often used in the form of grated cheese, applied as a sprinkled ingredient on "pasta italiana, bolognese, etc.", how should the properties of hard cheese be evaluated from the point of view of this treatment?

8. Hard cheeses produced in Italy are very popular and are available from different regions and, according to available information, they also differ in certain details in the production technology. Based on experience from other areas of food evaluation, the habit of the evaluator towards a certain food product (food, drink) also plays a certain role in the evaluation. I would like to know the opinion of the authors on the evaluation and comparison of Parmigiano Reggiano cheese with other similar cheeses, e.g. Grana Padano, Gran Biraghi, Gran Moravia, and other Italian or foreign hard cheeses with similar production technology.

Round 2

Reviewer 1 Report

Comments and Suggestions for Authors

Comments to the Authors

The manuscript untitled “Consumer perception and liking of Parmigiano Reggiano PDO cheese produced with milk from cows fed fresh forage”, manuscript ID foods-2809868 - Revised Version, contains interesting information and could contribute to scientific advances. After analysis of the revised manuscript, I notice that it improved a lot, and the author made an afford to insert all the suggestion given by the reviewers. 

I notice that the manuscript is written according to the rules of the journal and the scientific issues are well explained and organized. 

Just one suggestion, that was proposed, and the authors accept it (see the authors´s answer) but they didn´t change in in the manuscript:

Results and discussion 

Precise, well written and organized.
However, I suggest you change the title of point 3.1.
Answer: We thank the reviewer for this comment; we modified this title in the revised version of the manuscript accordingly. 

Due to the comments above, it is my opinion that the manuscript could be accepted with minor revisions.

Author Response

Results and discussion 

Precise, well written and organized.
However, I suggest you change the title of point 3.1.
Answer: We thank the reviewer for this comment; we modified this title in the revised version of the manuscript accordingly. 

Due to the comments above, it is my opinion that the manuscript could be accepted with minor revisions.

Answer: We thank the reviewer for highlight this mistake. We thought that section 3.1 was the socio-demographic one but, due to an error in numbering of the section, also the 3.2 was numbered as 3.1, the section related to liking. Thus, we modified the title of this section in the revised version from “Liking” to “Hedonic assessment of Parmigiano Reggiano samples” (highlighted in green colour).
